# Implementing Rounding Checklists in a Pediatric Oncologic Intensive Care Unit

**DOI:** 10.3390/children9040580

**Published:** 2022-04-18

**Authors:** Mary Katherine Gardner, Patricia J. Amado, Muhummad Usman Baig, Sana Mohiuddin, Avis Harden, Linette J. Ewing, Shehla Razvi, Jose A. Cortes, Rodrigo Mejia, Demetrios Petropoulos, Priti Tewari, Ali H. Ahmad

**Affiliations:** 1Pediatric Critical Care, Division of Nursing, The University of Texas MD Anderson Cancer Center, Houston, TX 77030, USA; mkgardner@mdanderson.org (M.K.G.); pjcraven@mdanderson.org (P.J.A.); 2Pediatric Oncology Fellowship Program, Department of Pediatrics, The University of Texas MD Anderson Cancer Center, Houston, TX 77030, USA; mubaig@llu.edu (M.U.B.); sana.mohiuddin@hhsys.org (S.M.); aharden@mdanderson.org (A.H.); 3Section of Pediatric Critical Care, Department of Pediatrics, The University of Texas MD Anderson Cancer Center, Houston, TX 77030, USA; ljewing@mdanderson.org (L.J.E.); srazvi@mdanderson.org (S.R.); jacortes@mdanderson.org (J.A.C.); rmejia@mdanderson.org (R.M.); 4Section of Pediatric Stem Cell Transplantation and Cellular Therapy, Department of Pediatrics, The University of Texas MD Anderson Cancer Center, Houston, TX 77030, USA; dpetro@mdanderson.org (D.P.); ptewari@mdanderson.org (P.T.)

**Keywords:** pediatrics, cancer, oncology, checklists, rounds, critical care

## Abstract

Standardized rounding checklists during multidisciplinary rounds (MDR) can reduce medical errors and decrease length of pediatric intensive care unit (PICU) and hospital stay. We added a standardized process for MDR in our oncologic PICU. Our study was a quality improvement initiative, utilizing a four-stage Plan–Do–Study–Act (PDSA) model to standardize MDR in our PICU over 3 months, from January 2020 to March 2020. We distributed surveys to PICU RNs to assess their understanding regarding communication during MDR. We created a standardized rounding checklist that addressed key elements during MDR. Safety event reports before and after implementation of our initiative were retrospectively reviewed to assess our initiative’s impact on safety events. Our intervention increased standardization of PICU MDR from 0% to 70% over three months, from January 2020 to March 2020. We sustained a rate of zero for CLABSI, CAUTI, and VAP during the 12-month period prior to, during, and post-intervention. Implementation of a standardized rounding checklist may improve closed-loop communication amongst the healthcare team, facilitate dialogue between patients’ families and the healthcare team, and reduce safety events. Additional staffing for resource RNs, who assist with high acuity patients, has also facilitated bedside RN participation in MDR, without interruptions in clinical care.

## 1. Introduction

Accurate communication is an essential process within healthcare teams in the intensive care setting. Sentinel events and medical errors are frequently related to a failure in communication [1,2,3,4,5]. Donchin et al. demonstrated that a failure to transfer the correct information resulted in 1.7 errors per patient per day [1]. Similarly, Narasimhan et al. found that medical errors are more likely to happen when miscommunication occurs [6].

Closed-loop communication is a method of ensuring information transmission. The keys steps involved in closed-loop communication include the sender delivering information, the receiver acknowledging receipt of information, and the receiver verbally confirming the information delivered [7,8].

Multidisciplinary rounds (MDR) are a widely implemented communication method of sharing information, evaluating patients, and formulating therapeutic plans, particularly useful in intensive care units (ICUs) [5,9]. MDR typically include physicians and bedside registered nurses (RNs), as well as respiratory therapists, pharmacists, case managers, child life specialists, ethicists, and patients’ families [10].

However, lack of a specific rounding structure, team members’ lack of clear understanding of patient care goals, and interruptions during rounds may contribute to miscommunication, and therefore clinical errors [11,12]. Other factors that may affect rounds are personnel with differing approaches in communication, change in personnel including residents, and the absence of key team members during rounds [13].

The use of a standardized rounding checklist by physicians and nurses during MDR can result in lower medical errors, decreased length of ICU stay, and decreased length of hospital stay [13,14,15,16,17]. Despite these benefits, implementation of standardized rounding checklists is inconsistent, as was the case in our pediatric oncologic ICU. Pediatric oncologic ICUs are care units uniquely subspecialized to care for critically ill children with cancer. The medical complexity and fragile physiology of critically ill pediatric oncology patients require attention to detail and a shared mental model.

Our pediatric oncologic ICU consists of nine beds, with an average age of 14 years for admitted patients and 175 annual admissions on average over the last five years. The average length of pediatric ICU (PICU) stay was 15.3 days over twelve months prior to our intervention. Reasons for PICU admission ranged from acute respiratory failure, septic shock, cytokine release syndrome, encephalopathy, electrolyte disturbances, and post-operative neurosurgical, orthopedic, vascular, and other subspecialty oncological surgery care, amongst other diagnoses. The average sequential organ failure assessment (SOFA) score for patients admitted to PICU was 3. We had thirteen PICU RNs at the beginning of the study and fourteen PICU RNs at the end of the study. Our PICU RNs worked 12-h shifts on either days or nights. The nurse-to-patient ratio would either be one-to-one or one-to-two. We had five attending pediatric intensivists who provided 24/7 in-house coverage, one daytime PICU nurse practitioner (NP), and two nighttime PICU physician assistants (PA) throughout the study. We also had one PICU fellow at a time, completing a one-month oncologic PICU rotation each month.

Before the implementation of our project, there was a great degree of variability in when rounds occurred, who was present at MDR, who would present the patient during MDR, who would make a clinical assessment of the patient, and who would develop a formal plan of care for the patient. The major barriers towards standardization that we identified included the lack of a specific rounding structure, staff intensivists with different training and practice styles, and the absence of key team members during rounds.

To address these issues, we implemented a SMART (Specific, Measurable, Achievable, Realistic, and Timely) aim with a goal to increase audit compliance, as a surrogate measure of MDR standardization, from 0% to 70% in the PICU over three months, from January 2020 to March 2020 [18]. The global aim was to improve team communication by the implementation of an RN-driven standardized rounding checklist. Our secondary aims were to assess bedside RN understanding and satisfaction regarding communication during MDR, reduce medical errors in the ICU, and assess length of ICU stay before and after intervention.

## 2. Materials and Methods

Our study was a quality improvement initiative at The University of Texas MD Anderson Cancer Center, utilizing a four-stage Plan–Do–Study–Act (PDSA) model. We created an MDR workflow that included bedside RN and family presence at MDR, as well as an organ system-based standardized rounding checklist that addressed key elements during MDR (Figure 1).

Before starting the PDSA cycles, we distributed an anonymous survey to pediatric ICU (PICU) RNs, who provide primary bedside care to patients in our ICU, to assess RN understanding and satisfaction regarding communication during MDR. Educational sessions, focused on standardized MDR and family-centered care, were delivered in the morning and evening by the study team. These sessions introduced the expectations for RN-driven standardized rounding checklist implementation and participation, and they also discussed the importance of family-centered care.

Our standardized MDR occurred twice a day—9 a.m. and 9 p.m. The healthcare staff present at standardized MDR consisted of the bedside PICU RN, PICU NP during the day (PICU PA at night), PICU fellow, and PICU attending. This team expanded when rounding with the pediatric stem cell transplant (SCT) team—which consisted of a pediatric SCT NP, pediatric SCT pharmacist, pediatric oncology fellow, and pediatric SCT attending. Night rounds would usually not include the pediatric SCT team. The role of the bedside RN was to present the patient using the checklist as displayed on the left in Figure 1. The PICU NP or PA would then make an assessment of the patient and begin the construction of a plan. The PICU fellow would then expand upon the assessment and make a detailed plan by organ systems, as displayed on the right in Figure 1. The PICU NP or PA would place orders in the electronic medical record in real time as the PICU fellow developed the plan. The PICU attending would then make a high-level summary of the patient, provide feedback and revisions as necessary to the assessment and plan, and engage a dialogue with the family to answer any questions about the clinical status of the patient and the treatment plan for the day. If the pediatric SCT team was present, they would address their team’s specific issues after the bedside PICU RN presented, but before the rest of the PICU team proceeded. The bedside RN would document the plan as displayed on the right in Figure 1, and a verbal readback would occur at the end of MDR for the patient to ensure closed-loop communication. This process would range from approximately 15 min to 30 min per patient, depending on the complexity of the patient and the time required to address the clinical and social concerns of each patient.

Following the implementation of the new nurse-driven standardized rounding checklist, reminder emails were sent to physicians and RNs to encourage its use. A follow-up survey with the same questions was distributed at the end of our study period.

The rounding checklist included necessary items to be addressed during MDR twice daily, organized by organ system. With each successive PDSA cycle, the RN champions for our study provided feedback to the study team regarding the efficiency and practicality of the rounding checklists. Each PDSA cycle was designated as 1 month in duration. Checklist revisions were made based upon feedback after each cycle. Three pediatric oncology fellows, members of our study team, but independent from the PICU clinical team, directly observed rounds, and performed “audits”, to assess key components of MDR at specific time points, at baseline, and after each PDSA cycle, to determine our initiative’s impact on the standardization of MDR. Each pediatric oncology fellow audited MDR multiple times during our three-month intervention period. The audits only occurred during morning rounds. A two-year follow-up series of audits was completed in March 2022 by our PICU NP. The PDSA cycle workflow is demonstrated in Figure 2.

Safety event reports from the six-month period prior to the implementation of our initiative (August 2019 to January 2020) and the six-month period following implementation of our initiative (March 2020–August 2020) were retrospectively reviewed to assess our initiative’s impact on safety events. There were four categories for the type of events reviewed: omission/error in assessment, medication related, laboratory tests, and care coordination/communication. An example of a laboratory test related safety issue included events such as a specimen being lost in transit to the laboratory. An example of a care coordination related safety issue included events such as incomplete patient handoff on admission to the PICU from other locations within hospital. Patient census and acuity were similar pre- and post-implementation. Safety event reporting culture had not changed during the twelve-month study period.

We also collected data on central line associated blood stream infection (CLABSI), catheter associated urinary tract infection (CAUTI), and ventilator associated pneumonia (VAP) during a 12-month period prior to, during and post-intervention (October 2019–September 2020). MDACC follows the National Healthcare Safety Network (NHSN) criteria for defining healthcare-associated infections. CLABSI was defined as a primary bloodstream infection (that is, there is no apparent infection at another site) that develops in a patient with a central line in place within the 48-h period before onset of the bloodstream infection that is not related to infection at another site. CAUTI was defined as a UTI where an indwelling urinary catheter was in place for more than two days. VAP was defined as pneumonia that occurs at least 48–72 h following endotracheal intubation, characterized by the presence of a new or progressive infiltrate, signs of systemic infection (fever, altered white blood cell count), changes in sputum characteristics, and detection of a causative agent.

## 3. Results

### 3.1. Nurse Surveys

A seven-question survey was distributed to bedside PICU RNs before and after our quality improvement initiative, which is detailed in Figure 3. Five out of thirteen RNs (38.5%) responded to the pre-intervention survey. Nine out of fourteen RNs (64.3%) responded to the post-intervention survey.

### 3.2. PDSA Cycle Data

Figure 4 details the PDSA cycle results. At baseline during MDR, there was *n* = 0/8 (0%) participation from the charge RN, *n* = 8/8 (100%) participation from the bedside RN, time provided for the bedside RN to report the plan of the day was provided *n* = 6/8 (75%), all components of the rounding checklist were addressed *n* = 0/8 (0%) of the time, and family was invited to attend rounds *n* = 2/8 (25%) of the time. Our audit compliance was 40% at baseline.

Following the initial nursing survey and education sessions, three PDSA cycles were assessed. For PDSA Cycle 1, charge RN participation during rounds improved to *n* = 3/5 (60%), bedside RN participation was *n* = 5/5 (100%), time provided for the bedside RN to report the plan was *n* = 3/5 (60%), all checklist components addressed improved to *n* = 3/5 (80%), and family invitation improved to *n* = 2/4 (50%). Our audit compliance was 70% for Cycle 1.

During Cycle 2, charge RN participation was *n* = 4/7 (57%), bedside RN participation remained at *n* = 7/7 (100%), time provided for the bedside RN to report the plan increased to *n* = 7/7 (100%), all components of the checklist were addressed *n* = 6/7 (86%) of the time, and family was invited to participate in rounds *n* = 1/4 (25%) of the time. Our audit compliance was 78% for Cycle 2.

During Cycle 3 charge RN participation was 0%, bedside RN participation was *n* = 3/4 (75%), time provided for the bedside RN to report the plan was *n* = 4/4 100%, all components of the checklist were addressed *n* = 4/4 (100%), and family was invited to participate *n* = 2/4 (50%) of the time. Our audit compliance was 70% for Cycle 3.

A two-year follow-up was completed in March 2022, titled Cycle 4 in Figure 4. Charge RN participation was *n* = 3/5 (60%), bedside RN participation was *n* = 5/5 (100%), time provided for the bedside RN to report the plan was *n* = 3/4 (100%), all components of the checklist were addressed *n* = 5/5 (100%) of the time, and family was invited to participate *n* = 3/3 (100%) of the time. Our audit compliance was 92% for Cycle 4.

### 3.3. Safety Event Reports

Figure 5 details the safety event data. Before our intervention, there were seven safety event reports related to omission/error in assessment, and post-implementation there were two events reported in this category. There were seven medication-related safety event reports both pre-and post-implementation of our quality initiative. Pre-intervention, there were thirty laboratory test safety event reports, and this decreased to sixteen events post-intervention. Lastly, there were four care coordination/communication event reports pre-intervention and three of these reports post-intervention.

### 3.4. CLABSI, CAUTI, and VAP Data

Appendix A detail CLABSI, CAUTI, and VAP results. There were 1222 central line days, 312 urinary catheter days, and 123 ventilator days in our pediatric oncologic ICU during this time period. We sustained a rate of zero for CLABSI, CAUTI, and VAP during the 12-month period prior to, during and post-intervention (October 2019–September 2020).

## 4. Discussion

We demonstrated an improvement in the standardization of PICU MDR during this initiative, and we were also able to improve RN perception of communication. Safety event reports either remained at baseline or decreased after our initiative. By incorporating our initiative into our daily workflow, we facilitated closed-loop communication on our patients’ plan of care amongst our team members and encouraged dialogue between patients’ families and the medical team. As a result, we continue to use our RN-driven rounding workflow and checklist for every patient admitted to our pediatric oncologic ICU. Since this initiative, we have also established a new role of resource RN, who can assist with bedside care in high acuity patients, and also facilitate bedside RN participation in MDR, without interruptions in clinical care.

The PICU is a complex and busy environment, where care for a critically ill child includes frequent assessments and numerous diagnostic studies, medications, and procedures [4,14]. The aviation industry has a two-decade history of managing an analogous environment by utilizing checklists, where reliance on memory may lead to errors [19,20,21]. Thus, we adapted a checklist model to standardize our MDR process based upon a successful approach in prior studies [14,19].

Improved communication leads to improved patient outcomes [22,23]. For example, Pronovost et al. found that implementing a daily goals form in a surgical oncology ICU improved nurse understanding of goals of care for the day, and reduction in ICU length of stay [22]. Thus, we prioritized nursing perception of communication as one of our study outcomes in our pediatric oncologic ICU.

We chose an RN-driven checklist, as the bedside RN is often the member of the healthcare team who spends the most time with the patient and the family. As such, they are uniquely positioned to serve as the best patient advocate and communication bridge between patients and their families with the rest of the healthcare team [14,23].

We included specific elements to audit each PDSA cycle based upon previous literature by Tufnaru et al., who issued best practice recommendations to standardize MDR by the use of a consistent MDR start time, the presence of bedside RN and family during MDR, empowering all MDR participants to share their perspective for patient care, and a structured daily goals checklist [24]. Charge RN support substantially increased during our initiative from 0% to about 60% from baseline through Cycle 2. The drop-off displayed during Cycle 3 correlated to March 2020, as we experienced an increased clinical demand across disciplines due to the COVID-19 pandemic. As expected, an already 100% bedside RN participation at baseline remained stable throughout our initiative until Cycle 3, where we experienced a similar drop-off. The time allotted for the RN to review the plan of day for the patient, and all items on our MDR checklist being addressed steadily rose throughout our initiative. Finally, while family invitation to participate in MDR was variable, the PICU team always communicated with patients and families after MDR regardless of their presence during MDR.

There was a high degree of variability in RN survey responses prior to our initiative, and a significant increase in consistency amongst the responses post-initiative. This reflects our study’s influence on the standardization of timing and attendance of MDR, improvement on the efficiency and effectiveness of MDR, and clarity of communication during MDR. Interestingly, previous investigators noted organizations that improved RN perception of nurse–physician collaboration were able to reduce nursing turnover [25].

Ultimately, the importance of daily goal sheets, checklists, and the standardization of care is to ensure patient safety. Pronovost and Holzmueller described that repetition of information decreases the likelihood of error [26]. Thus, we added safety event reports as one of our study outcomes. Although safety event reports either remained at baseline or decreased after our initiative, it is important to note that these reports are self-reported by employees, so this is a convenience sample rather than a true safety event rate. It is entirely possible that safety events occurred prior to, during, or after our intervention that were not recorded in the data that we have reported.

Similarly, CLABSI, CAUTI, and VAP rates were at zero before and after our initiative, so it is difficult to assess what affect our initiative had on these findings. Of note, the majority of our patients have surgically implanted semi-permanent central venous ports for chemotherapy infusion. Thus, our RN staff have a great degree of familiarity with central line care. Additionally, our standardized rounding checklist included specific verbiage about central line dressing changes. Furthermore, our relatively low urinary catheter and ventilator days may contribute to our zero CAUTI and VAP rates. For our surgical patients, our center has published an Enhanced Recovery Program which aims to standardize perioperative treatment plans for pediatric oncologic surgical patients, including, but not limited to, early mobility which often involves liberation from medical devices [27].

There are several limitations to this study. Firstly, the timeframe of our project was limited in duration. The second limitation was the advent of the COVID-19 pandemic. This crisis diverted staff and attention towards more emergent needs at the time. However, we were still able to see the project through to completion and maintain a high level of compliance. Third, the method of self-reported safety event reports may have omitted other safety events that were not detected by our study. Fourth, our survey results reflected the opinions of those nurses who chose to complete the survey, thus opening the risk for selection bias with these results. Similarly, the time at which the audits occurred was based upon the auditors’ availability. The auditors were pediatric oncology fellows and were not part of the PICU team, making them independent auditors. Although we attempted to have the fellows stagger their audits every week, to audit each PICU attending during their service week, at times the audits occurred during the same week, which also could lead to a selection bias. These audits occurred exclusively during morning rounds, which also predisposes a selection bias by omitting compliance assessments during night rounds. Fifth, participation of family members remained low throughout the study. This seems to be an area for potential improvement as the family was not regularly invited to participate in MDR. Occasionally, this was due to mitigating circumstances, such as a patient being on airborne isolation precautions. The family would be updated by the PICU attending after MDR in these cases. At the beginning of 2020, pre-pandemic, our visitation policy allowed for both parents to visit all day and to board overnight. Once the pandemic began, one parent was allowed to board with the patient, but the parent had to stay in the hospital indefinitely without leaving. The policy evolved throughout the course of the pandemic, depending on caseloads and new variants arising. After the first wave of the pandemic, the policy was relaxed so that the one parent would board for a week at a time, at which point the parents could swap visitation. Our most recent visitation policy allows for both parents visit all day and to board overnight. If a patient is nearing end of life, then exceptions are made to allow for other family members to visit. This may have also affected the results.

Finally, it is noteworthy that in the time since this intervention, we have had significant PICU RN turnover despite an improvement in RN perception of nurse–physician communication. This is in the context of a nationwide nurse staffing shortage seen after a two-year pandemic that has driven many ICU nurses to pursue a variety of other employment opportunities [28]. Despite these changes, our initiative has fostered a culture of standardization and our checklist has helped our new PICU RN staff acclimate to their new environment. Further studies will explore the sustainability of these interventions and the impact on patient-centered outcomes.

## 5. Conclusions

The PICU is a complex and busy environment, where care for a critically ill child includes numerous assessments, diagnostic studies, medications, and procedures. Implementation of a standardized rounding checklist may improve closed-loop communication amongst the interdisciplinary healthcare team, facilitate dialogue between patients’ families and the healthcare team, and reduce safety events.

## Figures and Tables

**Figure 1 children-09-00580-f001:**
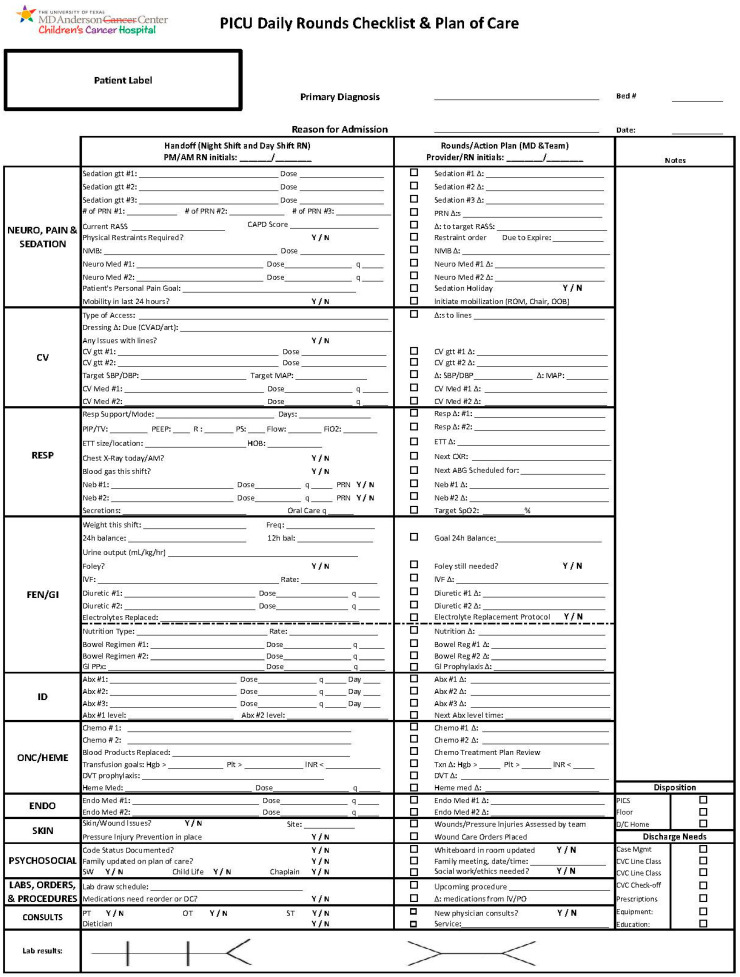
Pediatric Intensive Care Unit Multidisciplinary Rounds (MDR) Checklist. Organ system-based standardized rounding checklist that addressed key elements during MDR. Organ systems listed in first column from left, patient data presented from second column, patient plan of care for shift created in third column, and additional notes annotated in fourth column. PICU = Pediatric Intensive Care Unit. NEURO = Neurologic. gtt = Gutta (continuous infusion). PRN = Pro Re Nata (as needed). RASS = Richmond Agitation–Sedation Scale. CAPD = Cornell Assessment for Pediatric Delirium. NMB = Neuromuscular blockade. q = Every. ROM = Range of Motion. OOB = Out of Bed. CV = Cardiovascular. CVAD = Cardiovascular Access Device. SBP = Systolic Blood Pressure. DBP = Diastolic Blood Pressure. MAP = Mean Arterial Pressure. RESP = Respiratory. PIP = Peak Inspiratory Pressure. TV = Tidal Volume. PEEP = Positive End-Expiratory Pressure. R = Rate. PS = Pressure Support. FiO2 = Fraction of Inspired Oxygen. ETT = Endotracheal Tube. HOB = Head of Bed. CXR = Chest X-ray. ABG = Arterial Blood Gas. SpO2 = Saturation of Peripheral Oxygen. FEN = Fluids, Electrolytes, and Nutrition. GI = Gastrointestinal. Freq = Frequency. Bal = Balance. IVF = Intravenous Fluids. PPx = Prophylaxis. ID = Infectious Diseases. Abx = Antibiotics. Onc = Oncology. Heme = Hematology. Chemo = Chemotherapy. Hgb = Hemoglobin. Plt = Platelets. INR = International Normalized Ratio. DVT = Deep Venous Thrombosis. Txn = Transfusion. ENDO = Endocrine. SW = Social Work. DC, D/C = Discontinue. IV = Intravenous. PO = Per os (oral). PT = Physical Therapy. OT = Occupational Therapy. ST = Speech Therapy. PICS = Pediatric Intensive Care Service. CVC = Central Venous Catheter.

**Figure 2 children-09-00580-f002:**
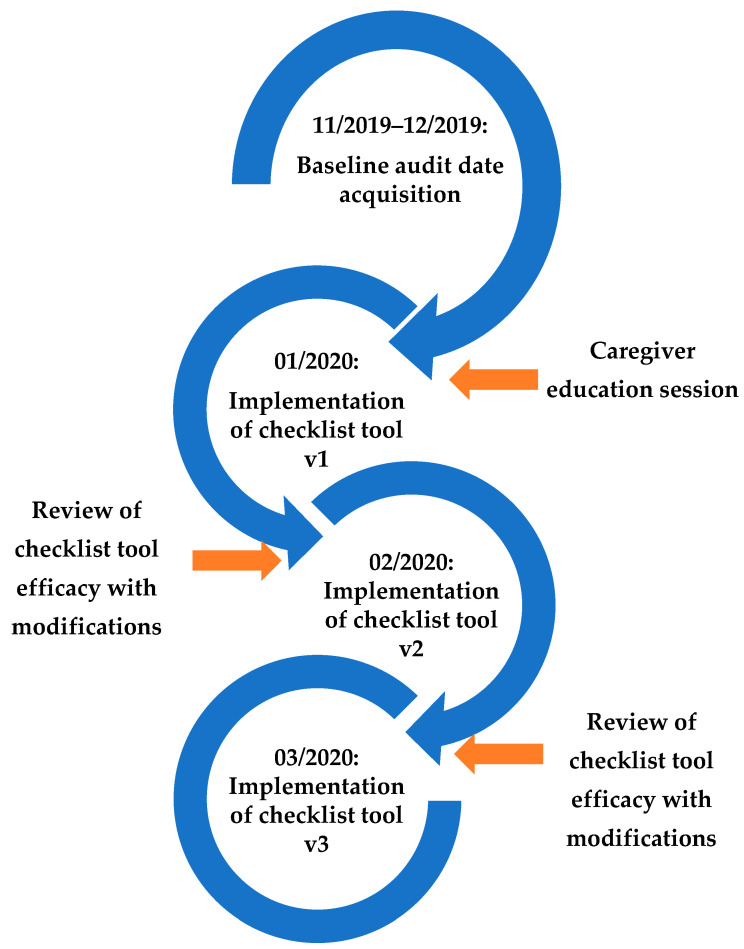
Plan–Do–Study–Act (PDSA) cycle workflow. With each successive PDSA cycle, the RN champions for our study provided feedback to the study team regarding the efficiency and practicality of the rounding checklists. Each PDSA cycle was designated as 1 month in duration. Checklist revisions were made based upon feedback after each cycle. v1 = Version 1. v2 = Version 2. v3 = Version 3.

**Figure 3 children-09-00580-f003:**
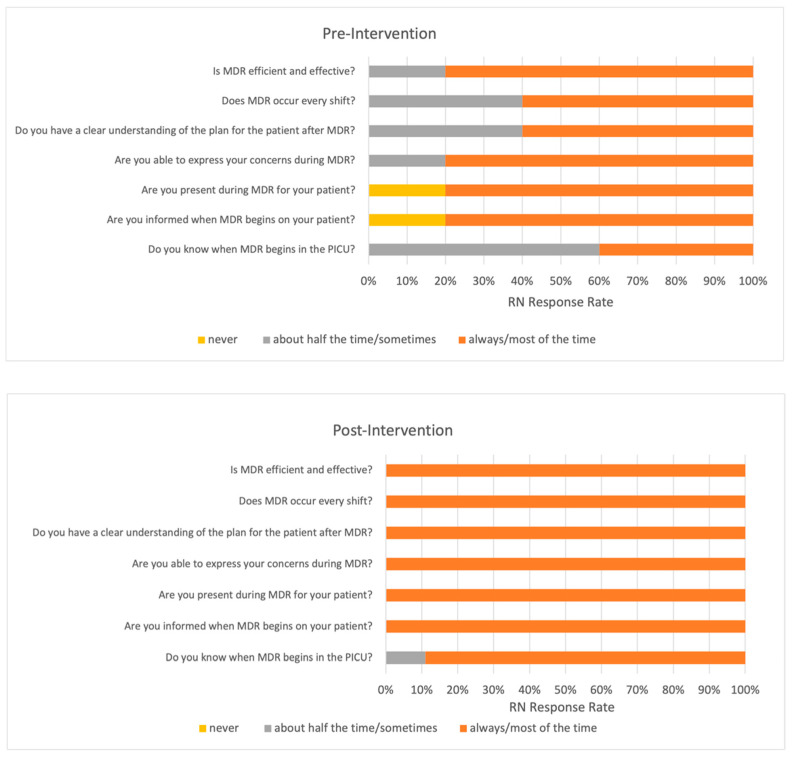
PICU Registered Nurse Survey Results. MDR = Multidisciplinary Rounds. A seven-question survey was distributed to bedside PICU RNs before and after our quality improvement initiative. Five out of thirteen RNs (38.5%) responded to the pre-intervention survey. Nine out of fourteen RNs (64.3%) responded to the post-intervention survey. PICU = Pediatric Intensive Care Unit. RN = Registered Nurse.

**Figure 4 children-09-00580-f004:**
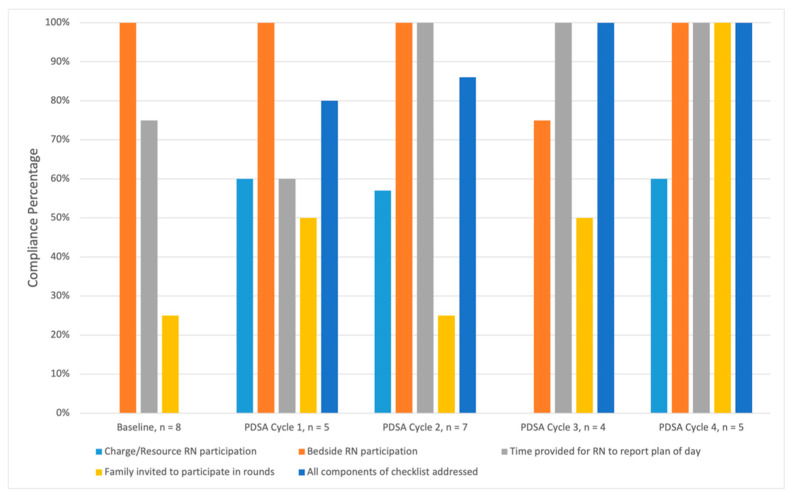
Plan–Do–Study–Act Cycle Data. Pediatric Intensive Care Unit multidisciplinary rounds were observed by study team members to assess for key elements listed in legend. *y*-axis demonstrates compliance percentage for each element during a cycle. *x*-axis demonstrates each element conducted in cycles over time. Baseline data were acquired from November 2019–December 2019. Cycle 1 data were acquired January 2020. Cycle 2 data were acquired February 2020. Cycle 3 data were acquired March 2020. A two-year follow-up was completed in March 2022, listed as Cycle 4. PDSA = Plan–Do–Study–Act.

**Figure 5 children-09-00580-f005:**
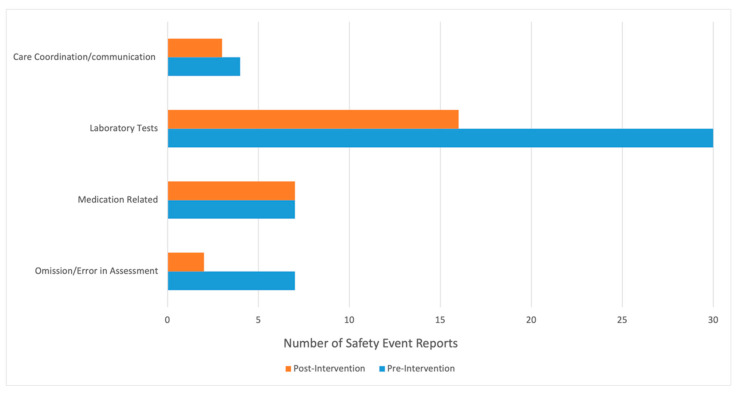
Safety Event Data. Safety event reports from the six-month period prior to implementation of our initiative (August 2019 to January 2020) and the six-month period following implementation of our initiative (March 2020–August 2020) were retrospectively reviewed to assess our initiative’s impact on safety events. *y*-axis demonstrates categories for the type of events reviewed. *x*-axis demonstrates the number of safety event reports for each category, with blue bars representing pre-intervention and orange bars representing post-intervention.

## Data Availability

The data presented in this study are available on request from the corresponding author.

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
