# Peer review of "Implementing Rounding Checklists in a Pediatric Oncologic Intensive Care Unit"

_children, 2022, doi:10.3390/children9040580_

Round 1

Reviewer 1 Report

I would add in the introduction a description of the Pediatric Oncologic Intensive Care Unit, with the number of beds, average age hospitalized and annual number of hospitalizations. The number members participating to the daily MDR and the the number of physicians and RNs is also interesting.

I am not clear in figures 6-8 where a zero rate is claimed for CLABSI, CAUTI and VAP. The number of ventilatory days is low and therefore we have zero VAP, but in oncological PICU I would expect a CLABSI RATE higher than that of the Pediatric cohort.

Author Response

I would add in the introduction a description of the Pediatric Oncologic Intensive Care Unit, with the number of beds, average age hospitalized and annual number of hospitalizations. The number members participating to the daily MDR and the the number of physicians and RNs is also interesting.

Thank you for your thoughtful review and comments. We added additional verbiage that states:

  1. Our pediatric oncologic ICU consists of nine beds, with an average age of 14 years for admitted patients and 175 annual admissions on average over the last five years. The average length of pediatric ICU (PICU) stay was 15.3 days over twelve months prior to our intervention. Reasons for PICU admission ranged from acute respiratory failure, septic shock, cytokine release syndrome, encephalopathy, electrolyte disturbances, and post-operative neurosurgical, orthopedic, vascular, and other subspecialty oncological surgery care, amongst other diagnoses. The average sequential organ failure assessment (SOFA) score for patients admitted to PICU was 3. We had thirteen PICU RNs at the beginning of the study and fourteen PICU RNs at the end of the study. Our PICU RNs worked 12-hour shifts on either days or nights. The nurse-to-patient ratio would either be one-to-one or one-to-two. We had five attending pediatric intensivists who provided 24/7 in-house coverage, one daytime PICU nurse practitioner (NP), two nighttime PICU physician assistants (PA) throughout the study. We also had one PICU fellow at a time, completing a one-month oncologic PICU rotation each month. Before the implementation of our project, there was a great degree of variability in when rounds occurred, who was present at MDR, who would present the patient during MDR, who would make a clinical assessment of the patient, and who would develop a formal plan of care for the patient.
  2. Our standardized MDR occurred twice a day – 9 AM and 9 PM. The healthcare staff present at standardized MDR consisted of the bedside PICU RN, PICU NP during day (PICU PA at night), PICU fellow, and PICU attending. This team expanded when rounding with the pediatric stem cell transplant (SCT) team – which consisted of a pediatric SCT NP, pediatric SCT pharmacist, pediatric oncology fellow, and pediatric SCT attending. Night rounds would usually not include pediatric SCT team. The role of the bedside RN was to present the patient using the checklist as displayed on the left in Figure 1. The PICU NP or PA would then make an assessment of the patient and begin the construction of a plan. The PICU fellow would then expand upon the assessment and make detailed plan by organ systems as displayed on the right in Figure 1. The PICU NP or PA would place orders in the electronic medical record in real time as the PICU fellow developed the plan. The PICU attending would then make a high-level summary of the patient, provide feedback and revisions as necessary to the assessment and plan, and engage a dialogue with the family to answer any questions about the clinical status of the patient and the treatment plan for the day. If the pediatric SCT team was present, they would address their team’s specific issues after the bedside PICU RN presented, but before the rest of the PICU team proceeded. The bedside RN would document the plan as displayed on the right in Figure 1, and a verbal readback would occur at the end of MDR for the patient to ensure closed-loop communication.

I am not clear in figures 6-8 where a zero rate is claimed for CLABSI, CAUTI and VAP. The number of ventilatory days is low and therefore we have zero VAP, but in oncological PICU I would expect a CLABSI RATE higher than that of the Pediatric cohort. 

We added additional verbiage that states: 

  1. MDACC follows the National Healthcare Safety Network (NHSN) criteria for defining healthcare-associated infections. CLABSI was defined as a primary bloodstream infection (that is, there is no apparent infection at another site) that develops in a patient with a central line in place within the 48-hour period before onset of the bloodstream infection that is not related to infection at another site. CAUTI was defined as UTI where an indwelling urinary catheter was in place for more than two days. VAP was defined as pneumonia that occurs at least 48-72 hours following endotracheal intubation, characterized by the presence of a new or progressive infiltrate, signs of systemic infection (fever, altered white blood cell count), changes in sputum characteristics, and detection of a causative agent.
  2. There were 1,222 central line days, 312 urinary catheter days, and 123 ventilator days in our pediatric oncologic ICU during this time period. Figure 6-8 detail CLABSI, CAUTI and VAP results. We sustained a rate of zero for CLABSI, CAUTI and VAP during the 12-month period prior to, during and post-intervention (October 2019-September 2020).
  3. Similarly, CLABSI, CAUTI, and VAP rates were at zero before and after our initiative, so it is difficult to assess what affect our initiative had on these findings. Of note, the majority of our patients have surgically implanted semi-permanent central venous ports for chemotherapy infusion. Thus, our RN staff have a great degree of familiarity with central line care. Additionally, our standardized rounding checklist included specific verbiage about central line dressing changes. Furthermore, our relatively low urinary catheter and ventilator days may contribute to our zero CAUTI and VAP rates. For our surgical patients, our center has published an Enhanced Recovery Program which aims to standardize perioperative treatment plans for pediatric oncologic surgical patients, including, but not limited to, early mobility which often involves liberation from medical devices [25].
  1. Wells SJ, Austin M, Gottumukkala V, Kruse B, Mayon L, Kapoor R, Lewis V, Kelly D, Penny A, Braveman B, Shkedy E, Crowder R, Moody K, Swartz MC. Development of an Enhanced Recovery Program in Pediatric, Adolescent, and Young Adult Surgical Oncology Patients. Children (Basel). 2021 Dec 8;8(12):1154. doi: 10.3390/children8121154. PMID: 34943351; PMCID: PMC8700533.

Reviewer 2 Report

Comments to the authors

This study aims to propose the use of standardized rounding checklists during MDRs as an effective initiative to improve communication among the healthcare team members, and thus improving clinical outcomes of the critically ill pediatric patients. The authors implemented standardized rounding checklists during MDRs over three months and reported improved outcomes in many areas such as PDSA cycle, safety event reports, CLABSI, CAUTI, VAP, and nurse survey. By demonstrating the efficacy of the standardized rounding checklists, this paper suggests that healthcare team could improve clinical outcomes by drastically reducing miscommunication errors among the team members. I greatly appreciate the value and potential of this paper but at the same time have several concerns regarding the ways in which the authors conducted research and reported the results, which I detailed below.

  1. More specific and detailed information regarding the MDRs of your team would be desirable. For example, what was the occupational composition of your team, and how many people from each occupational group participated in MDRs? How often was MDR? How regularly? What was the role of the person from each occupational group? How was the rotation schedule of each occupation? How many people (as a total and/or from each occupational group) participated in this study? Were those people allowed to participate multiple times over the 3-month periods? Were there any correlations between the repeated number of participation and the performance?
  2. Were there any differences between morning and night rounds? Did you check if the results of your study differ by the timing of MDRs?
  3. How many auditors (i.e., the PICU fellows) did participate in this study? Was there any auditors who assessed more than once? Do the results of the study change if you take this into account?
  4. Only bedside RNs and family members used the standardized rounding checklists? Then, what was the role of other occupational people such as residents or fellows during MDRs?
  5. The mention of the following details would be useful, too. How many beds in your pediatric oncologic ICU? What is the ratio of nurse to patient? How about the nurse shifting? Did family members reside always? If not, on average, how many hours per day, and how many people were with the patients? How often? Were they allowed to be with patients even during the corona pandemic period? If not, how different was the rule before and after corona pandemic period? These parameters might have affected the results.
  6. What did you mean by the care coordination or laboratory test related safety issue? Can you explain more specifically? If possible, explanation with examples would be great.
  7. The CLABSI, CAUTI, and VAP results (i.e., sustaining a rate of zero) is very impressive, especially as considering that the data was from the pediatric oncologic ICU!, What were the diagnostic criteria used? Did you use any specialized control methods? Then, what do you think about the relationship between using standardized rounding checklists and zero of CLABSI, CAUTI, and VAP? There were no item about instrument insertion and/or removal, duration, dressing/bathing methods or compatibility assessment, etc. in the standardized rounding checklist. Do you think that there are any boundary conditions or moderating variables between the relationship between standardized rounding checklists and CLABSI, CAUTI, and VAP? Please discuss these in the discussion section.
  8. Mechanical vent day was relatively low. Can you explain the characteristics of the enrolled patients and this ICU? For example, cause of PICU admission, the severity of illness, and duration of PICU day, etc.
  9. Did you implement standardized rounding checklists only for 3 months? Or you implemented checklists more than 3 months but did audit for the study only for 3 months? Given that it is already March 2022, do you have any follow-up data related to implementation of standardized rounding checklists?
  10. It seems to me that participation of the family members and nurse survey were quite low. What do you think are the reasons behind this low participation rate?

Author Response

This study aims to propose the use of standardized rounding checklists during MDRs as an effective initiative to improve communication among the healthcare team members, and thus improving clinical outcomes of the critically ill pediatric patients. The authors implemented standardized rounding checklists during MDRs over three months and reported improved outcomes in many areas such as PDSA cycle, safety event reports, CLABSI, CAUTI, VAP, and nurse survey. By demonstrating the efficacy of the standardized rounding checklists, this paper suggests that healthcare team could improve clinical outcomes by drastically reducing miscommunication errors among the team members. I greatly appreciate the value and potential of this paper but at the same time have several concerns regarding the ways in which the authors conducted research and reported the results, which I detailed below.

1. More specific and detailed information regarding the MDRs of your team would be desirable. For example, what was the occupational composition of your team, and how many people from each occupational group participated in MDRs? How often was MDR? How regularly? What was the role of the person from each occupational group? How was the rotation schedule of each occupation? How many people (as a total and/or from each occupational group) participated in this study? Were those people allowed to participate multiple times over the 3-month periods? Were there any correlations between the repeated number of participation and the performance?

Thank you for your thoughtful review and comments. We added additional verbiage that states:

  1. We had thirteen PICU RNs at the beginning of the study and fourteen PICU RNs at the end of the study. Our PICU RNs worked 12-hour shifts on either days or nights. The nurse-to-patient ratio would either be one-to-one or one-to-two. We had five attending pediatric intensivists who provided 24/7 in-house coverage, one day-time PICU nurse practitioner (NP), two nighttime physician assistants (PA) throughout the study. We also had one PICU fellow completing a one-month oncologic PICU rotation each month. Before the implementation of our project, there was a great degree of variability in when rounds occurred, who was present at MDR, who would present the patient during MDR, who would make a clinical assessment of the patient, and who would develop a formal plan of care for the patient.
  2. Our standardized MDR occurred twice a day – 9 AM and 9 PM. The healthcare staff present at standardized MDR consisted of the bedside PICU RN, PICU NP during day (PICU PA at night), PICU fellow, and PICU attending. This team expanded when rounding with the pediatric stem cell transplant (SCT) team – which consisted of a pediatric SCT NP, pediatric SCT pharmacist, pediatric oncology fellow, and pediatric SCT attending. Night rounds would usually not include pediatric SCT team. The role of the bedside RN was to present the patient using the checklist as displayed on the left in Figure 1. The PICU NP or PA would then make an assessment of the patient and begin the construction of a plan. The PICU fellow would then expand upon the assessment and make detailed plan by organ systems as displayed on the right in Figure 1. The PICU NP or PA would place orders in the electronic medical record in real time as the PICU fellow developed the plan. The PICU attending would then make a high-level summary of the patient, provide feedback and revisions as necessary to the assessment and plan, and engage a dialogue with the family to answer any questions about the clinical status of the patient and the treatment plan for the day. If the pediatric SCT team was present, they would address their team’s specific issues after the bedside PICU RN presented, but before the rest of the PICU team proceeded. The bedside RN would document the plan as displayed on the right in Figure 1, and a verbal readback would occur at the end of MDR for the patient to ensure closed-loop communication.
  3. Three pediatric oncology fellows, members of our study team, but independent from the PICU clinical team, directly observed rounds, and performed “audits”, to assess key components of MDR at specific time points, at baseline, and after each PDSA cycle, to determine our initiative’s impact on standardization of MDR. Each pediatric oncology fellow audited MDR multiple times during our three-month intervention period. The audits only occurred during morning rounds.

2. Were there any differences between morning and night rounds? Did you check if the results of your study differ by the timing of MDRs?

We added additional verbiage that states: 

  1. Three pediatric oncology fellows, members of our study team, but independent from the PICU clinical team, directly observed rounds, and performed “audits”, to assess key components of MDR at specific time points, at baseline, and after each PDSA cycle, to determine our initiative’s impact on standardization of MDR. Each pediatric oncology fellow audited MDR multiple times during our three-month intervention period. The audits only occurred during morning rounds.
  2. Similarly, the time at which the audits occurred was based upon the auditors’ availability. The auditors were pediatric oncology fellows and were not part of the PICU team, making them independent auditors. Although we attempted to have the fellows stagger their audits every week to audit each PICU attending during their service week, at times the audits occurred during the same week, which also could lead to a selection bias. These audits occurred during morning rounds, which also predisposes a selection bias by omitting compliance assessments during night rounds.

3. How many auditors (i.e., the PICU fellows) did participate in this study? Was there any auditors who assessed more than once? Do the results of the study change if you take this into account?

We added additional verbiage that states: 

  1. Three pediatric oncology fellows, members of our study team, but independent from the PICU clinical team, directly observed rounds, and performed “audits”, to assess key components of MDR at specific time points, at baseline, and after each PDSA cycle, to determine our initiative’s impact on standardization of MDR. Each pediatric oncology fellow audited MDR multiple times during our three-month intervention period. The audits only occurred during morning rounds.
  2. Similarly, the time at which the audits occurred was based upon the auditors’ availability. The auditors were pediatric oncology fellows and were not part of the PICU team, making them independent auditors. Although we attempted to have the fellows stagger their audits every week to audit each PICU attending during their service week, at times the audits occurred during the same week, which also could lead to a selection bias. These audits occurred during morning rounds, which also predisposes a selection bias by omitting compliance assessments during night rounds.

4. Only bedside RNs and family members used the standardized rounding checklists? Then, what was the role of other occupational people such as residents or fellows during MDRs?

We added additional verbiage that states: 

  1. The role of the bedside RN was to present the patient using the checklist as displayed on the left in Figure 1. The PICU NP or PA would then make an assessment of the patient and begin the construction of a plan. The PICU fellow would then expand upon the assessment and make detailed plan by organ systems as displayed on the right in Figure 1. The PICU NP or PA would place orders in the electronic medical record in real time as the PICU fellow developed the plan. The PICU attending would then make a high-level summary of the patient, provide feedback and revisions as necessary to the assessment and plan, and engage a dialogue with the family to answer any questions about the clinical status of the patient and the treatment plan for the day. If the pediatric SCT team was present, they would address their team’s specific issues after the bedside PICU RN presented, but before the rest of the PICU team proceeded. The bedside RN would document the plan as displayed on the right in Figure 1, and a verbal readback would occur at the end of MDR for the patient to ensure closed-loop communication.
  2. We chose an RN-driven checklist, as the bedside RN is often the member of the healthcare team who spends the most time with the patient and the family. As such, they are uniquely positioned to serve as the best patient advocate and communication bridge between patients and their families with the rest of the healthcare team.

5. The mention of the following details would be useful, too. How many beds in your pediatric oncologic ICU? What is the ratio of nurse to patient? How about the nurse shifting? Did family members reside always? If not, on average, how many hours per day, and how many people were with the patients? How often? Were they allowed to be with patients even during the corona pandemic period? If not, how different was the rule before and after corona pandemic period? These parameters might have affected the results.

We added additional verbiage that states: 

    1. Our pediatric oncologic ICU consists of nine beds, with an average age of 14 years for admitted patients and 175 annual admissions on average over the last five years. We had thirteen PICU RNs at the be-ginning of the study and fourteen PICU RNs at the end of the study. Our PICU RNs worked 12-hour shifts on either days or nights. The nurse-to-patient ratio would either be one-to-one or one-to-two.
    2. At the beginning of 2020, pre-pandemic, our visitation policy allowed for both parents to visit all day and to board overnight. Once the pandemic began, one parent was allowed to board with the patient, but the parent had to stay in the hospital indefinitely without leaving. The policy evolved throughout the course of the pandemic, depending on caseloads and new variants arising. After the first wave of the pandemic, the policy was relaxed so that the one parent would board for a week at a time, at which point the parents could swap visitation. Our most recent visitation policy allows for both parents visit all day and to board overnight. If a patient is nearing end of life, then exceptions are made to allow for other family members to visit. This may have also affected the results.

6. What did you mean by the care coordination or laboratory test related safety issue? Can you explain more specifically? If possible, explanation with examples would be great.

We added additional verbiage that states: 

    1. There were four categories for the type of events reviewed: omission/error in assessment, medication related, laboratory tests, and care coordination/communication. An example of laboratory test related safety issues included events such as a specimen being lost in transit to the laboratory. An example of care coordination related safety issued included events such as incomplete patient handoff on admission to PICU from other locations within hospital.

7. The CLABSI, CAUTI, and VAP results (i.e., sustaining a rate of zero) is very impressive, especially as considering that the data was from the pediatric oncologic ICU!, What were the diagnostic criteria used? Did you use any specialized control methods? Then, what do you think about the relationship between using standardized rounding checklists and zero of CLABSI, CAUTI, and VAP? There were no item about instrument insertion and/or removal, duration, dressing/bathing methods or compatibility assessment, etc. in the standardized rounding checklist. Do you think that there are any boundary conditions or moderating variables between the relationship between standardized rounding checklists and CLABSI, CAUTI, and VAP? Please discuss these in the discussion section.

We added additional verbiage that states: 

    1. MDACC follows the National Healthcare Safety Network (NHSN) criteria for defining healthcare-associated infections. CLABSI was defined as a primary bloodstream infection (that is, there is no apparent infection at another site) that develops in a patient with a central line in place within the 48-hour period before onset of the bloodstream infection that is not related to infection at another site. CAUTI was defined as UTI where an indwelling urinary catheter was in place for more than two days. VAP was defined as pneumonia that occurs at least 48-72 hours following endotracheal intubation, characterized by the presence of a new or progressive infiltrate, signs of systemic infection (fever, altered white blood cell count), changes in sputum characteristics, and detection of a causative agent.
    2. There were 1,222 central line days, 312 urinary catheter days, and 123 ventilator days in our pediatric oncologic ICU during this time period. Figure 6-8 detail CLABSI, CAUTI and VAP results. We sustained a rate of zero for CLABSI, CAUTI and VAP during the 12-month period prior to, during and post-intervention (October 2019-September 2020).
    3. Similarly, CLABSI, CAUTI, and VAP rates were at zero before and after our initiative, so it is difficult to assess what affect our initiative had on these findings. Of note, the majority of our patients have surgically implanted semi-permanent central venous ports for chemotherapy infusion. Thus, our RN staff have a great degree of familiarity with central line care. Additionally, our standardized rounding checklist included specific verbiage about central line dressing changes. Furthermore, our relatively low urinary catheter and ventilator days may contribute to our zero CAUTI and VAP rates. For our surgical patients, our center has published an Enhanced Recovery Program which aims to standardize perioperative treatment plans for pediatric oncologic surgical patients, including, but not limited to, early mobility which often involves liberation from medical devices [25].

8. Mechanical vent day was relatively low. Can you explain the characteristics of the enrolled patients and this ICU? For example, cause of PICU admission, the severity of illness, and duration of PICU day, etc.

We added additional verbiage that states: 

    1. The average length of PICU stay was 15.3 days over twelve months prior to our intervention. Reasons for PICU admission ranged from acute respiratory failure, septic shock, cytokine release syndrome, encephalopathy, electrolyte disturbances, and post-operative neurosurgical, orthopedic, vascular, and other subspecialty oncological surgery care, amongst other diagnoses. The average sequential organ failure assessment (SOFA) score for patients admitted to PICU was 3.

9. Did you implement standardized rounding checklists only for 3 months? Or you implemented checklists more than 3 months but did audit for the study only for 3 months? Given that it is already March 2022, do you have any follow-up data related to implementation of standardized rounding checklists?

  1. We conducted audits for 3 months, but standardized MDR and checklist have remained a part of our PICU.

We also added additional verbiage that states: 

b. Finally, it is noteworthy that in the time since this intervention, we have had significant PICU RN turnover despite an improvement in RN perception of nurse-physician communication. This is in the context of a nationwide nurse staffing shortage seen after a two-year pandemic that has driven many ICU nurses to pursue a variety of other employment opportunities [26]. Despite these changes, our initiative has fostered a culture of standardization and our checklist has helped our new PICU RN staff acclimate to their new environment. Further studies will explore the sustainability of these interventions and the impact on patient-centered outcomes.

10. It seems to me that participation of the family members and nurse survey were quite low. What do you think are the reasons behind this low participation rate?

The following verbiage is included in the manuscript:

  1. The second limitation was the advent of the coronavirus-19 pandemic. This crisis diverted staff and attention towards more emergent needs at the time. […] Fourth, our survey results reflected the opinions of those nurses who chose to complete the survey, thus opening the risk for selection bias with these results.

We also added additional verbiage that states: 

b. Fifth, participation of family members remained low throughout the study. This seems to be an area for potential improvement as the family was not regularly invited to participate in MDR. Occasionally, this is due to mitigating circumstances, such as a patient being on airborne isolation precautions. The family would be updated by the PICU attending after MDR in these cases.

  1. Wells SJ, Austin M, Gottumukkala V, Kruse B, Mayon L, Kapoor R, Lewis V, Kelly D, Penny A, Braveman B, Shkedy E, Crowder R, Moody K, Swartz MC. Development of an Enhanced Recovery Program in Pediatric, Adolescent, and Young Adult Surgical Oncology Patients. Children (Basel). 2021 Dec 8;8(12):1154. doi: 10.3390/children8121154. PMID: 34943351; PMCID: PMC8700533.
  2. Wahlster S, Sharma M, Lewis AK, Patel PV, Hartog CS, Jannotta G, Blissitt P, Kross EK, Kassebaum NJ, Greer DM, Curtis JR, Creutzfeldt CJ. The Coronavirus Disease 2019 Pandemic's Effect on Critical Care Resources and Health-Care Providers: A Global Survey. Chest. 2021 Feb;159(2):619-633. doi: 10.1016/j.chest.2020.09.070. Epub 2020 Sep 11. PMID: 32926870; PMCID: PMC7484703.